# Identification and Expression of Integrins during Testicular Fusion in *Spodoptera litura*

**DOI:** 10.3390/genes14071452

**Published:** 2023-07-15

**Authors:** Yaqing Chen, Yu Chen, Baozhu Jian, Qili Feng, Lin Liu

**Affiliations:** 1Guangdong Provincial Key Laboratory of Insect Developmental Biology and Applied Technology, Institute of Insect Science and Technology, School of Life Sciences, South China Normal University, Guangzhou 510631, China; ccyaqing@163.com (Y.C.); 2020022797@m.scnu.edu.cn (Y.C.); 2019200473@m.scnu.edu.cn (B.J.); qlfeng@scnu.edu.cn (Q.F.); 2Guangzhou Key Laboratory of Insect Development Regulation and Application Research, Institute of Insect Science and Technology, School of Life Sciences, South China Normal University, Guangzhou 510631, China

**Keywords:** integrin, testis, testicular fusion, extracellular matrix, *Spodoptera litura*

## Abstract

Integrin members are cell adhesion receptors that bind to extracellular matrix (ECM) proteins to regulate cell–cell adhesion and cell-ECM adhesion. This process is essential for tissue development and organogenesis. The fusion of two testes is a physiological phenomenon that is required for sperm production and effective reproduction in many Lepidoptera. However, the molecular mechanism of testicular fusion is unclear. In *Spodoptera litura*, two separated testes fuse into a single testis during the larva-to-pupa transformation. We identified five α and five β integrin subunits that were closely associated with testicular fusion. Integrin α1 and α2 belong to the position-specific 1 (PS1) and PS2 groups, respectively. Integrin α3, αPS1/αPS2, and αPS3 were clustered into the PS3 group. Integrin β1 belonged to the insect β group, and β2, β3, and β5 were clustered in the βν group. Among these integrins, α1, α2, α3, αPS1/PS2, αPS3, β1, and β4 subunits were highly expressed when the testes fused. However, their expression levels were much lower before and after the fusion of the testis. The qRT-PCR and immunohistochemistry analyses indicated that integrin β1 mRNA and the protein were highly expressed in the peritoneal sheath of the testis, particularly when the testes fused. These results indicate that integrins might participate in *S. litura* testicular fusion.

## 1. Introduction

Integrins are present in all metazoan animals, and they are cell adhesion receptors formed by the non-covalent association of α and β subunits. Each subunit consists of a type I transmembrane glycoprotein, a large extracellular domain, and a short cytoplasmic tail [1,2]. The crystal structure of integrin has a “bent” conformation that allows ligand binding near the membrane surface [3]. Integrin ligands, including the main ECM proteins like collagen, laminin, fibronectin, and vitronectin, have evolved with integrins [4,5]. By binding ligands, integrins transduce signals through the cell membrane in a bi-directional transmission (inside-to-outside or outside-to-inside signaling). This is critical for cell–cell or cell-ECM adhesion in tissue development and organogenesis in many-celled organisms [6,7,8,9].

In mammals, 18α and 8β subunits have been reported and assembled into 24 distinct integrin heterodimers with different ligand binding characteristics and different tissue distributions [1,10]. As species complexity increases, the number of integrins in the genome also increases [1,6]. Different expressions of integrin subunits depend on the cell type [11]. For example, the integrin β6 subunit is specifically expressed in epithelial cells [12,13] and plays key roles in regulating cell adhesion and the migration of cancer disease [14,15]. During the early gonadogenesis of mice, the primordial germ cells specifically expressed integrin beta3 [16] and played roles in cell adhesion during the theca cell layer formation in mouse ovarian follicles [17]. In the mouse ovary, integrin beta5 mRNA was primarily expressed in the granulosa cells of follicles [18,19], which suggested that integrin could play a function in granulosa cell proliferation. The specific expression patterns of integrin subunits in different cells and tissues are closely associated with their conservative functions in mediating cell–cell or cell-ECM adhesion.

Integrins have been shown to perform vital reproductive activities, including gamete formation, fertilization, and implantation [20,21]. In mouse spermatozoa, integrin α6/β4 were localized in the inner apical acrosomal membrane, integrin α3, α6/β1, and β4 were localized in the plasma membrane that covered the tip of the acrosome, and integrin α3/β1 were localized in the outer membrane of the acrosome [22]. In the rat testis, the α9 subunit was only found in the basement membrane and peritubular cells, while the α1 subunit was expressed in peritubular cells and the lamina propria. The α5 subunit was expressed in the acrosomes of elongating spermatids, as well as in their distal cytoplasm [23]. In *Drosophila*, it has been shown that integrin mediating cell-ECM adhesion can play important roles in muscle-tendon attachment at myotendinous junctions of the muscle tissue during larval development [24]. During *Drosophila*’s early oogenesis, integrin can medicate cell–basement membrane interactions to regulate epithelial cell division and differentiation [25].

Integrin expression in Lepidoptera demonstrates considerable tissue specialization. For example, in *Bombyx mori*, αPS3 is specifically expressed in granulosa cells [26]. These expression patterns of β integrins suggest that there may be a link between integrin expression and gonadal morphogenesis. Integrin β2 and β3 were expressed in the plasmatocytes of *B. mori* hemocytes and regulated the development of plasmatocytes [27]. In *Spodoptera exigua*, the reduction in β1 integrin by RNAi can cause a marked loss of cell–cell contacts and a remarkable midgut epithelium cell death when the Cry toxin infects the pests [28]. In *Helicoverpa armigera*, β integrin could be seen as a hemocytic receptor of C-type lectin during the encapsulation reaction analogous to granuloma formation in vertebrates [29]. In *Ostrinia furnacalis*, β1 integrin was expressed in the hemocytes and regulated the spreading of plasmatocytes [30]. These studies suggest that β1 integrin plays cell–cell adhesion and migration in the tissue development of the insect.

However, there are still fewer integrin subunits identified in insects. Seven integrin subunits including αPS1, αPS2, αPS3, αPS4, αPS5, βPS, and βν were identified in *D. melanogaste*r [31] and six α and five β integrin subunits were reported in *B. mori* [32]. *S. litura* (*S. litura*) (Lepidoptera: Noctuidae) is a major agricultural pest of many crops in the tropical and subtropical areas of Asia [33]. The *S. litura* testis produce spermatozoa and provides nutrition for spermatozoa development [34]. Testicular fusion occurs in many Lepidoptera [35,36,37]. In *S. litura*, the two kidney-like testes are separated during the larval stages. However, during their transition from final instar larva to pupa the two separated testes fuse to form a single testis [35,36,37,38]. The number of sperm bundles decreases if testicular fusion is artificially blocked by microsurgery. This finding indicates that testicular fusion is beneficial for sperm production [39]. However, the molecular mechanism of testicular fusion is still unknown. RNA-seq results have indicated that some integrin subunits may be involved in testicular fusion [38].

In this study, we identified cell adhesion receptor integrins of *S. litura* and analyzed their expression in the testis. The results confirmed the specific expression and localization of integrin subunits in the fusion of testis in *S. litura* and provided new insights into the molecular mechanism of testicular fusion in lepidopteran insects.

## 2. Materials and Methods

### 2.1. Experimental Insects

*S. litura* larvae were provided by the Entomology Institute of SUN YAT-SEN University (Guangzhou, China) and fed on an artificial diet, including soybean powder [40], wheat bran, and yeast powder [20]. These larvae matured into adults and laid eggs; the larvae were reared in an incubator at 26 ± 1 °C, with a 65 ± 5% relative humidity and a 14:10 h (L:D) photoperiod. The testes from 12 males at days 0, 2, 4, and 6 of the sixth instar larval stage (L6D0, L6D2, L6D4, and L6D6, respectively), as well as days 1 and 3 of the pupal stage (PD1 and PD3, respectively), were collected and washed in PBS for three times, were immediately frozen with liquid nitrogen, and stored at −80 °C until use as outlined in our previous description [38].

### 2.2. Prediction and Bioinformatic Analysis of Integrin Subunits

The integrin α and β subunits were predicted and analyzed, based on the published *S. litura* genome [41], using the Basic Local Alignment Search Tool (BLAST+) (http://ftp.ncbi.nlm.nih.gov/blast/executables/blast+/LATEST/) (accessed on 14 March 2022). The amino acid sequences of integrins from *Homo sapiens*, *Drosophila melanogaster*, *Bombyx mori*, *Tribolium castaneum*, *Manduca sexta*, *Pseudoplusia includens,* et al., were downloaded from NCBI (https://www.ncbi.nlm.nih.gov/, accessed on 14 March 2022) as reference sequences. The integrin sequences were obtained by querying the reference sequences to BLAST against the *S. litura* genome with the E-value threshold of 10^−6^ [42,43]. The putative integrins in *S. litura* were further verified by domain prediction using three online forecasting tools: SMART (http://smart.embl-heidelberg.de/, accessed on 14 March 2022), Pfam (http://pfam.xfam.org/, accessed on 14 March 2022) and PROSITE (https://prosite.expasy.org/, accessed on 14 March 2022).

### 2.3. Phylogeny Analysis of Integrins

The amino acid sequences of integrins from *H. sapiens*, *D. melanogaster*, *B. mori*, *S. litura*, and other species were aligned using the CLUSTAL W program [44]. The phylogenetic trees of integrin α and β subunits were constructed by the neighbor-joining method [45] with 1000 bootstrap replicates using the MEGA 6.0 program [46]. The gene bank accession numbers of all integrins are listed in Appendix A.

### 2.4. RNA Extraction, cDNA Synthesis, and Quantitative Real-Time PCR

The total RNA was extracted using the Trizol Reagent Kit (Invitrogen, Carlsbad, CA, USA) according to the manufacturer’s instructions. Two μg RNA from each sample was used for cDNA synthesis. The genomic DNA was digested using the RNase-free DNase I (TaKaRa, Dalian, China). The first-strand cDNA was synthesized using M-MLV Reverse Transcriptase (TaKaRa, Dalian, China) and oligo d(T) primers (TaKaRa, Dalian, China) using the manufacturer’s instructions. qRT-PCR reactions were carried out using the QuantStudio™ 6 Flex Real-Time system (ABI, Life Technologies, Carlsbad, CA, USA) with SYBR^®^ Select Master Mix (ABI). The reactions included a 10 μL of 2 × SYBR^®^ Select Master Mix (ABI) and 0.6 μL of specific forward and reverse primers (the concentration of each primer was 10 μM). The reactions were performed under the following conditions: 50 °C for 2 min, 95 °C for 2 min, followed by 40 cycles of 95 °C for 15 s and 60 °C for 1 min. The housekeeping gene *gapdh* (glyceraldehyde-3-phosphate dehydrogenase, LOC111366510) of *S. litura* was used as a reference gene [47]. The relative expression levels of these genes were analyzed using the method of 2^−ΔΔCt^ [48]. The primers of all the genes are listed in Appendix A. There were three repeats for all qRT-PCR reactions, and the experiment was repeated three times. One representative result is shown in the results.

### 2.5. Separation of Testicular Peritoneal Sheath and Sperm Cells

The testes from 12 individuals were collected for RNA extraction. The process for the separation of the peritoneal sheath and sperm cells was described previously [38]. The relative expression levels of integrin in the whole testis, peritoneal sheath, and sperm cells were tested by qRT-PCR and Western blotting. All experiments were repeated at least three times, and one representative result was shown.

### 2.6. Western Blotting Analysis

The total proteins from 20 testes were collected and extracted in a RIPA lysis buffer (Thermo Scientific, Guangzhou, China). A total of 25 μg of proteins from each sample were separated on 10% SDS-PAGE gels, and these proteins were then transferred onto PVDF membranes (Sigma-Aldrich, Shanghai, China) using a Trans-Blot Cell (Bio-Rad, NewYork, NY, USA). The primary antibody against *Sl*Integrin β1 was generated by immunized rabbits in the Zoonbio Biotechnology Company (Nanjing, China). The ratio of the primary antibody against *Sl*Integrin β1 (1:1000) was incubated at 4 °C overnight, and this membrane was washed three times with 0.1% Triton X-100 in phosphate-buffered saline (PBS, pH = 7.4, 140 mmol/L NaCl, 2.7 mmol/L KCl, 10 mmol/L Na_2_HPO_4_, 1.8 mmol/L KH_2_PO_4_). The HRP-conjugated goat anti-rabbit IgG (1:5000) was used as the second antibody and incubated at 37 °C for 1 h. Then, this membrane was washed three times with 0.1% Triton X-100 in PBS and twice with PBS alone. The rabbit antibody β-tubulin (Beyotime, Guangzhou, China) was used as a control reference. Using Image J (V1.8.0.112) (downloaded from https://imagej.nih.gov/ij/) (accessed on 1 May 2022), the relative expression of the Integrin β1 protein was estimated based on the integrated density ratio of Integrin β1 to β-tubulin in Western blot results.

### 2.7. Immunohistochemistry Analysis

The testes from L6D4, L6D6, and PD1 were fixed in 4% paraformaldehyde in PBS. The testes were dehydrated through a series of increasing concentrations of ethanol, which were cleared with xylene, and embedded in preheated melted paraffin at 60 °C, as described previously [49]. The paraffin sections (thickness = 5 μm) were prepared for immunohistochemistry. The slides were dewaxed and rehydrated for 5 min, respectively. Then, the slides were subjected to proteinase K treatment for 10 min, as previously described [50]. The anti-*Sl*Integrin antibody (a dilution of 1:50, in 2% BSA, 0.1% Tween 20 in PBS) was incubated for 1 h at 37 °C. The slides were washed three times at room temperature, 5 min at a time. The secondary antibody Goat anti-Rabbit IgG Alexa Fluor™ 594 (Invitrogen, Shanghai, China) (1:200) and DAPI (Beyotime, Shanghai, China) (1:2000) were incubated for 30 min at 37 °C. The slides were examined using a confocal microscope (Olympus Fluoview FV3000, Tokyo, Japan).

### 2.8. Statistical Analysis

Gene expression data were analyzed using GraphPad Prism 8 Software (GraphPad Software, San Diego, CA, USA), and significant differences were determined by ANOVA. The values provided are the mean ± SEM (standard error of the mean) (*n* = 3). A *p*-value < 0.05 or <0.01 indicated a significant difference.

## 3. Results

### 3.1. Identification of Integrin Members in S. litura

Genes representing members of the integrin family were identified by comparing the genomic sequences of *S. litura* and the published integrin sequences from *H. sapiens*, *D. melanogaster*, *B. mori*, and other species. A total of ten integrin members, five integrin α and five β subunits, were identified by searching the *S. litura* genome database (Table 1). The proteins of all of these integrins contained three domains: a large extracellular region, a single transmembrane region, and a short cytoplasmic region. This is consistent with the structure of the integrins found in other species [51].

### 3.2. Phylogenetic Analysis of Integrin Members

To study the relationship of integrin members in *S. litura* and other species, α and β phylogenetic trees were constructed using the full amino acid sequences from different species. *S. litura* integrin α subunits were divided into four groups (Figure 1), which is consistent with their previously reported evolutionary history [44]. The I-domain family, which possesses an “inserted” domain or “αI” domain clustered apart from others, has only been found in vertebrates (Figure 1). The PS1 and PS2 groups were found in both vertebrates and invertebrates, while the PS3 group was exclusively found in insects [5,32,44]. *S. litura* integrin α1 belonged to the PS1 group and was closely associated with *P. includens* integrin α1 (identity was 93.0%), *B. mori* integrin α1 (78.1%), and *D. melanogaster* integrin αPS1 (41.2%). *S. litura* Integrin α2 belonged to the PS2 group and was closely related to *P. includens* integrin α2 (79.4%), *M. sexta* integrin α2 (67.0%), and *B. mori* integrin α2 (57.8%). *S. litura* integrin α3, αPS1, αPS2, and αPS3 were clustered in the PS3 group, which contained *D. melanogaster* integrin αPS3, αPS4, and αPS5. *S. litura* integrin α3 had a 60.9% identity to *P. includens* integrin α3, *S. litura* αPS1 had a 32.1% identity to *B. mori* αPS2 (32.1%), and *S. litura* αPS2 had a 31.1% identity to *B. mori* αPS3 (Figure 1).

The integrin β subunits are mainly classified into five major phylogenetic groups, including two vertebrate groups: Vertebrate A (integrin β1, β2, β7) and Vertebrate B (integrin β3, β5, β6), and three insect β groups (Figure 2). Integrins in different insect orders, including Hymenoptera, Diptera, Coleoptera, and Lepidoptera, were included. *S. litura* integrin β1 was closely related to *S. exigua* integrin β1 (98.7%), and *S. litura* β2, β3, and β5 were clustered in the insect βν group. *S. litura* β4, *B. mori* β4, and *Danaus plexippus* β3 were clustered together in a single branch, which was different from the other groups. Interestingly, *S. litura* integrins αPS1/PS2 and αPS3 were located in the same scaffold of the *S. litura* genome, and the transcription orientation in the genome was the same, while integrin β2, β3, and β4 were also in the same scaffold; however, the transcriptional direction of β3 and β4 was inverse to that of integrin β2 (Appendix A).

### 3.3. Expression Patterns of Integrins in the Testis of S. litura at Different Developmental Stages

To study the expression patterns of the integrins identified in the testes, we collected the testes that were described in methods 2.1 for mRNA expression analysis by qRT-PCR. Integrin α1, α2, α3, αPS1/PS2, αPS3, β1, and β4 were highly expressed in L6D6 larvae when the testes were fusing and lower expressed in L6D0, L6D2, L6D4 larvae and PD1, PD3 pupae (Figure 3). Integrin β2 and β3 were highly expressed after the testicular fusion at PD3 pupae and were lower expressed in the other sixth larvae stages and PD1 pupa. The β5 had a high expression level in the new sixth instar after ecdysis (at L6D0), and its expression dropped significantly at L6D2 and gradually decreased from L6D4 to PD3 with the development of the testis (Figure 3). These results suggest that integrin subunits α1, α2, α3, αPS1/PS2, αPS3, β1, and β4 could participate in testicular fusion; however, integrin β2 and β3 subunits might be involved in testicular development after testicular fusion at the pupal stage. However, β5 could play a role in the early development of the testis at the larval stage.

### 3.4. Expression and Localization of Integrin β1 at Different Parts in the Testis of S. litura

Previous transcriptome analysis demonstrated the differential expression of integrin β1 during the testis fusion process [38]. To examine the protein expression and accurate location of integrin β1 in the fusing testes, an anti-integrin β1 polyclonal antibody was used to detect the expression and localization of integrin β1 in the testis. This peritoneal sheath was separated from sperm cells according to the method described previously [38]. The expression levels of integrin β1 mRNA detected by qRT-PCR and integrin β1 protein were detected by Western blotting and showed that the peritoneal sheath had a much higher expression than in the sperm cells around the fusion stage (Figure 4A,B). The expression and location of integrin β1 in the testis at the L6D4, L6D6, and PD1 stages were also determined by immunohistochemistry. The integrin β1 protein was specifically localized on the peritoneal sheath, particularly in the fusion side of the testis. This protein maintained a higher expression from L6D4 (before fusion) to L6D6 (on fusion) but dramatically decreased after testicular fusion at PD1 (Figure 5). These results suggest that integrin β1 is highly expressed in the peritoneal sheath when the testes are fusing and could participate in testicular fusion.

## 4. Discussion

In the present study, five α and five β integrin subunits were identified by aligning *S. litura* genome sequences with the reported integrin reference sequences from *B. mori*, *D.melanogaster*, and mammals. The sequences of the subunits αPS1 and αPS2 of *S. litura* were identical (Table 1). While in *B. mori*, αPS1, and αPS2 integrins were two different sequences [32]. The results implied that the integrin α subunit might have a different evolution between *B. mori* and *S. litura*. All members of the PS1, PS2, and PS3 groups appeared prior to the deuterostome and protostome divergence. The PS3 group underwent gene duplication in *Drosophila*, whereas the PS1 and PS2 groups underwent gene duplication only in vertebrates [5,44,52]. In *Drosophila*, the PS1 and PS2 integrin members participated in sperm–egg adhesion and fusion, which was accomplished by an interaction with the ligands on the sperm [22,53,54]. Together with ECM proteins, integrins promoted cell adhesion during the myoblast fusion, myotube formation process, and muscle, heart, and wing development [55,56,57,58]. In the *Drosophila* heart development process, the αPS3 integrin participated in lumen formation by leading cell migration and adhesion [59]. The PS3 integrin has been reported to be dispensable for dorsal appendage morphogenesis in epithelial cells of the *Drosophila* ovary. This suggests that redundant functions of integrins might be present within a simple tissue [60,61,62,63]. In the present study, all *S. litura* integrin α subunits were highly expressed when the testes were fusing at the L6D6 stage (Figure 3), indicating they might be involved in testicular fusion.

The *S. litura* integrin β1 might be conservative with other insects β1 by phylogenetic tree analysis. This is the first report to state that integrin β1 was mainly located on the perioneal sheath in the testis. Integrins β2, β3, and β5 could be conservative with the insect βν. In *S. exigua*, where the knockout of integrin β1 caused a significant loss in cell–cell contacts and cell death in the midgut epithelium, enhancing the efficacy of the *Bacillus thuringiensis* toxin [28]. Integrin β1 regulates cell migration and adhesion in the tracheal branches of *B. mori* [64]. In *M. sexta*, the binding of integrin β1 to the ECM ligand stimulates plasmatocyte spreading and adhesion, leading to encapsulation [65]. In the present study, based on the results of β integrins expression patterns, it was speculated that β1 and β4 integrins could participate in testicular fusion, while β2, β3, and β5 integrins might play a role before and after the fusion.

Integrin receptors connect to the cytoskeleton and bind to the ECM to mechanically link cellular components within and outside of the cells. We investigated the expression patterns of integrins during testicular fusion because it has not yet been established what function they play in gonadal morphogenesis in *S. litura*. We demonstrated that most of the integrin mRNAs in *S.litura* were highly expressed during testicular fusion. By studying the integrin expression pattern during the development of the *Drosophila* ovary, it was discovered that PS influenced the emergence of the egg chamber fusion [66]. The gene expression profile of β(1) integrin isoform b (ITGB1b) was quantified in the teleost fish gilthead seabream (*Sparus aurata* L., Teleostei) and analyzed in the context of the reproductive cycle, and it was found that the expression of ITGB1b was highly expressed during spermatogenesis, whereas the expression of other ECM-associated molecules was induced mainly in the post-spawning phase, suggesting that integrins were involved in the genesis of the reproductive system [67]. Therefore, we hypothesized that integrins could be involved in *S. litura* epithelial cell morphogenesis during the testicular fusion process, which is key to the construction of developing tissues and organs. For example, neural tube closure involves a process of epithelial fusion, whereby initially adjacent neural folds are adhered by cellular protrusions and then form a more stable union, which is followed by the remodeling of epithelial structures to form the neural tube [68]. This takes place in the closing of the optic fissure and eyelids [69]. In this study, the integrin β1 protein was highly expressed in the peritoneal sheath of the testes. Members of the integrin family played key roles as cell adhesion molecules in animal cell-to-cell and extracellular matrix interactions, such as integrin-mediated cell adhesion and spreading [70]. α6β4, an integrin that is expressed predominantly on the basal surface of most epithelial cells and a few other cell types, was found to be important for epithelial cell migration [71]. During *Drosophila* heart development, integrin-mediated cell adhesion and migration were essential for the formation of cardiac chambers, especially for cardioblast polarization [59,72]. This testicular fusion could be viewed to some extent as a fusion of the testes’ perithecal membrane and, during testicular fusion, could be accompanied by the migration of testis perithecal cells, which requires polarized membranes and cytoskeletal transport, polarization signals, and integrin signals. Thus, the high expression of integrin β1 in the testes’ perithecial membrane could be required to mediate cell migration and adhesion between two testes perithecial cells.

We previously reported [19] that the ECM remodeling of enzymes matrix metalloproteinases (*mmp2* and *mmp3*) and ECM proteins, including collagens, proteoglycans, and glycoproteins, were highly expressed in testicular ECM when testicular fusion occurred. During gonadal morphogenesis, integrin plays a variety of critical roles in cell migration, differentiation, and connection between the germ and somatic or somatic and somatic cells [16]. It is yet unknown how interactions and cellular signaling between these glycoproteins and integrins control gonadal morphogenesis. As a hint that there might be some connection between integrins and the testis’ fusion, our findings are expected to contribute to the research of testicular fusion in the *S. litura.*

## 5. Conclusions

In summary, this study demonstrated that all integrin α subunits and integrin β1, β4 had a higher expression during testicular fusion. Integrin β1 mRNA and the protein were highly expressed in the peritoneal sheath of the testes when the testes were in the process of fusion. Integrin β1, and other integrins, might be directly involved in the testicular fusion of *S. litura*. The molecular mechanisms involved in this process require further study.

## Figures and Tables

**Figure 1 genes-14-01452-f001:**
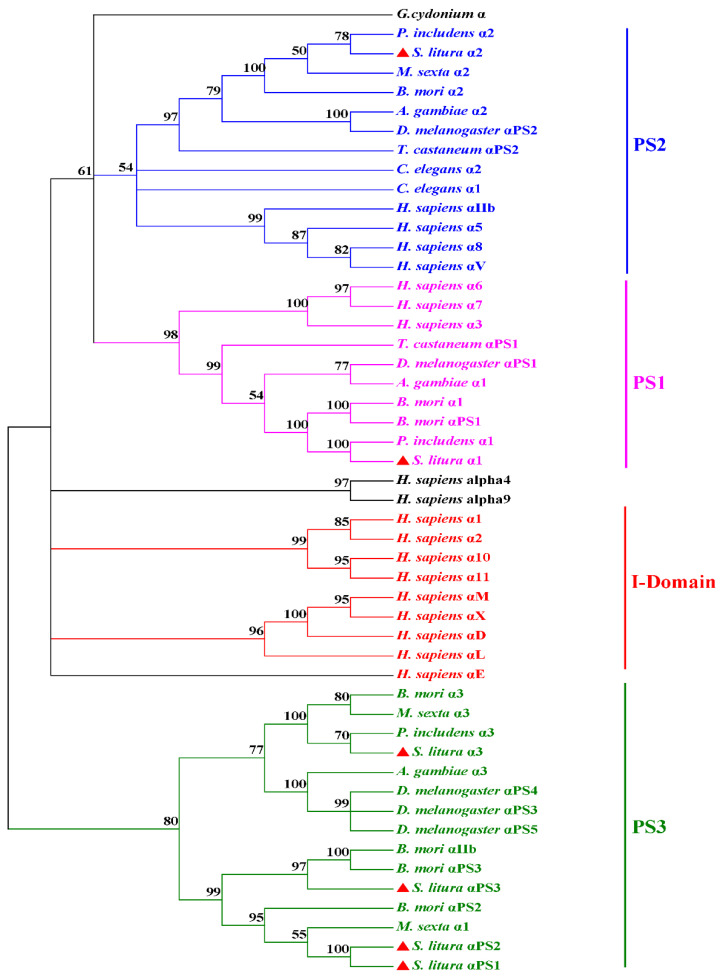
The phylogenetic tree of the integrin α family constructed by the neighbor-joining method and based on the full-length amino acid. The number on the branches represents the percentage of 1000 bootstrap iterations supporting the branch. Only the values above 50% are shown. The *S. litura* α subunits are labeled with triangles.

**Figure 2 genes-14-01452-f002:**
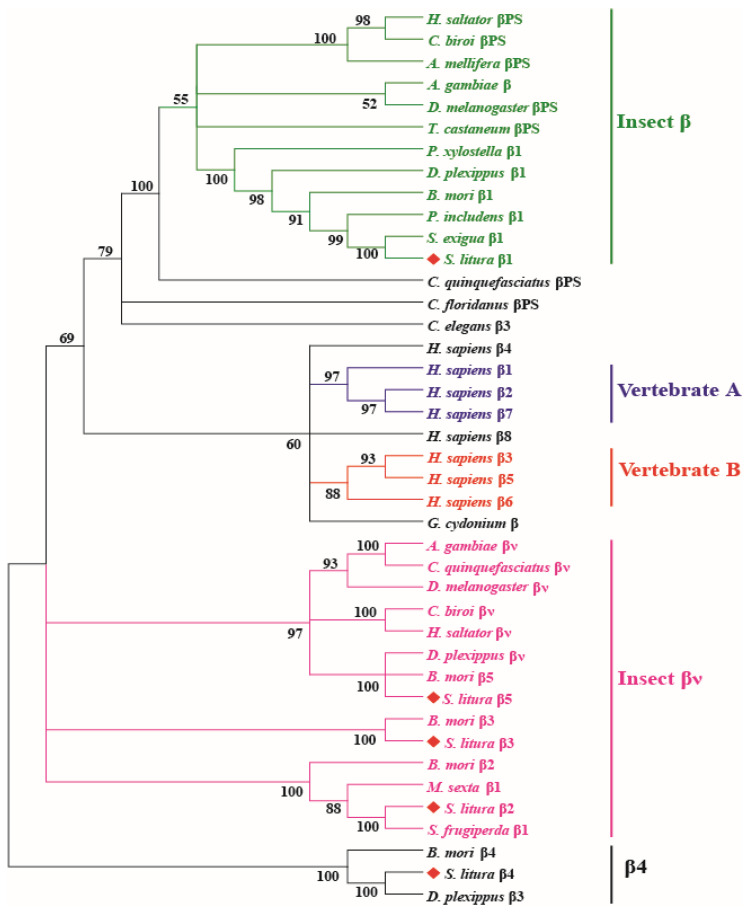
The phylogenetic tree of the integrin β family constructed by the neighbor-joining method and based on the full-length amino acid. The number on the branches represents the percentage of 1000 bootstrap iterations supporting the branch. Only the values above 50% are shown. The *S. litura* β subunits are labeled with rhombus shapes.

**Figure 3 genes-14-01452-f003:**
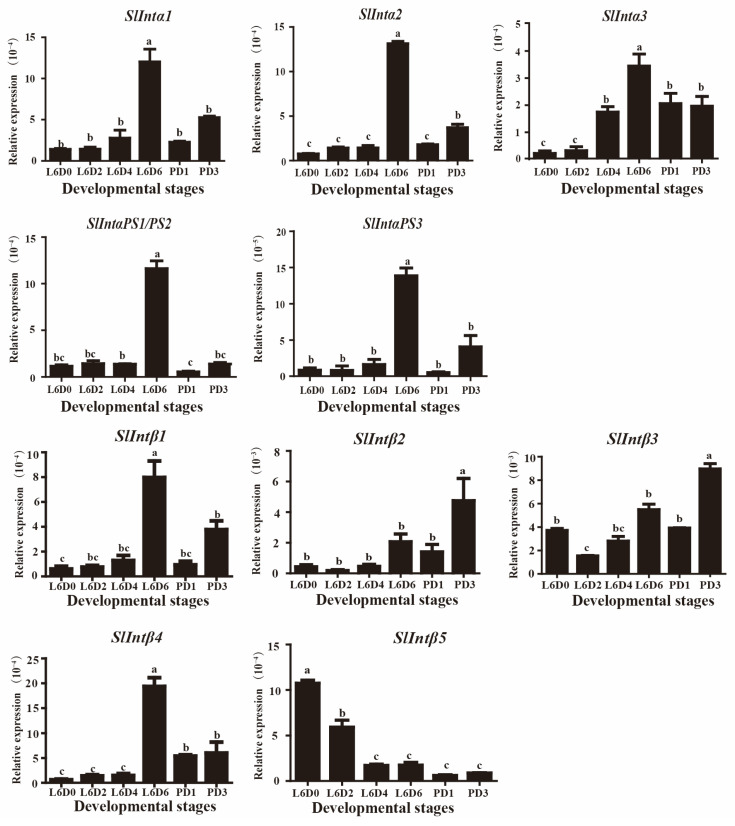
Expression pattern of integrin α subunits and β subunits in the testis at six developmental stages of *S. litura*. L6D0, L6D2, L6D4, and L6D6 represent days 0, 2, 4, and 6 of the sixth instar larval stage; P1D and P3D represent days 1 and 3 after pupation. The relative expression levels are normalized to the expression level of *Slgapdh*. The significance is indicated by different letters.

**Figure 4 genes-14-01452-f004:**
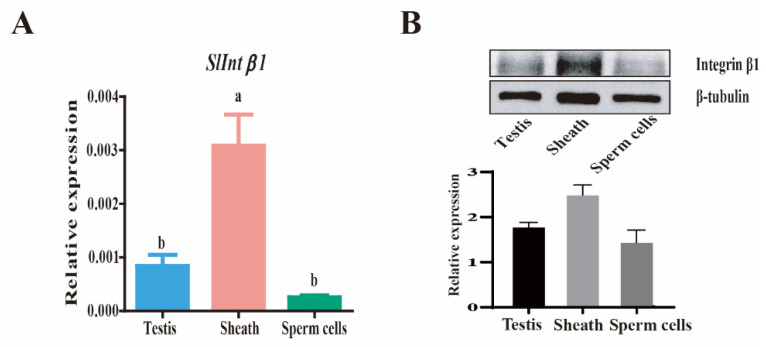
Expression pattern of the integrin β1 subunit in the testis, peritoneal sheath, and sperm cells at L6D6 determined by qRT-PCR (**A**) and Western blotting (**B**). The significance is indicated by different letters.

**Figure 5 genes-14-01452-f005:**
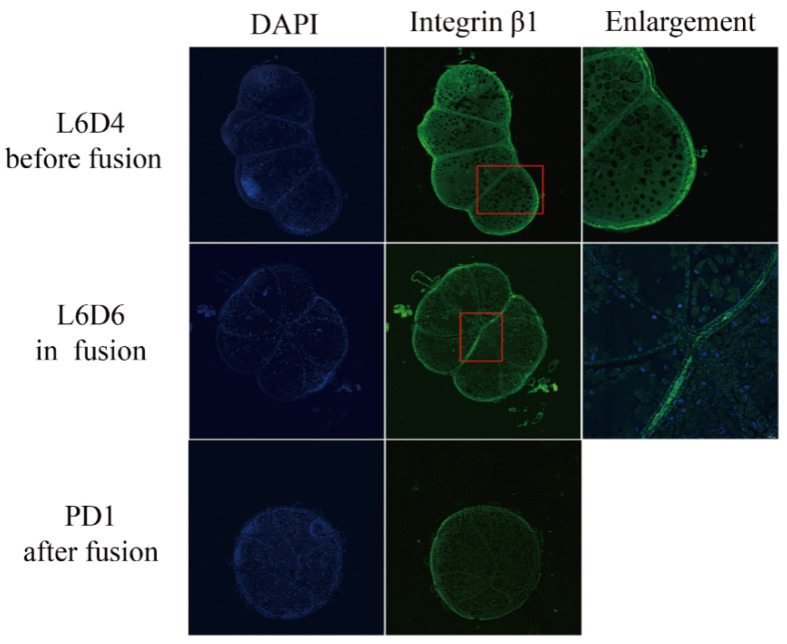
Expression and location of integrin β1 in the testis at L6D4 (before fusion), L6D6 (in fusion), and P1D (after fusion) by an immunofluorescence assay. L6D4 and L6D6 represent days 4 and 6 of the sixth instar larval stage; P1D represents 1 d after pupation. Red square is the area which image was amplified at the right side.

**Table 1 genes-14-01452-t001:** Summary of the integrin family identified in the *S. litura* genome.

Subunit	Gene	Scaffold	Position	Chr.	Accession	Exon	Size (AA)	MW/KDa
α	*α1*	scaffold157	623,547–706,210	6	XM_022982049	17	1118	124.21
	*α2*	scaffold285	13,930–127,259	14	XM_022963824	24	1567	174.05
	*α3*	scaffold417	212,791–229,830	7	XM_022959037	21	960	106.67
	*αPS1/PS2*	Scaffold384	257,012–280,655	14	XM_022964330	25	937	104.12
	*αPS3*	Scaffold384	217,418–233,535	14	XM_022964516	23	876	97.35
β	*β1*	scaffold315	223,689–230,408	14	XM_022963710	16	838	93.13
	*β2*	scaffold449(+)	44,554–48,972	15	XM_022964980	9	782	86.91
	*β3*	scaffold449(−)	237,386–243,026	15	XM_022965413	8	725	80.59
	*β4*	scaffold449(−)	247,062–275,746	15	XM_022965036	6	432	48.06
	*β5*	scaffold226	563,195–582,187	17	XM_022967414	13	807	89.69

## Data Availability

Not applicable.

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
