# Peer review of "Identification and Expression of Integrins during Testicular Fusion in Spodoptera litura"

_genes, 2023, doi:10.3390/genes14071452_

Round 1

Reviewer 1 Report

Reviewer comments:

The manuscript describes “Identification, expression, and localization of integrins during testes fusion in Spodoptera litura.  The title of the manuscript is wider than the subject incorporated in it. The author tries to understand the role of integrins during testes fusion using molecular and microscopic techniques. The manuscript is well written, which I overall accept. The parameters chosen for strategy building and informative.

Introduction: Need to be improved with essential points to be highlighted. Please extend this part to give more glimpse for understanding the subject and provide latest references.

Materials and methods: The structure and explanation of Section 2.1 can be improved.  Please provide all the information, unless you have already given an appropriate reference.

Please correct spelling "Testes" to Testis through out the manuscript.

Section 2.4: Do you try any other reference gene other than GAPDH? Incase yes, provide some information. It’s always better to use minimum two reference genes.

Results: Section 3.3 can be improved with more details since there is enough to explain.

Discussion part is very weak need to the reorganized because few parts of explanations are not connected and missing latest references. There should be a place where you need to explain the role of integrins explain mechanism that occur during developmental stage which is missing here.

General comments:

Please double-check the manuscript for errors. In fact, the chosen subject must be made explicit in the manuscript. The content presented here is more theoretical. There are many places that need to be updated with the latest reference. I suggest the authors to go for English corrections with a native speaker or a professional company.

If possible, please provide the graphical abstract. This can help readers to understand better.

Conclusion:

The manuscript is interesting but strategic connectivity to the proposed hypothesis is lacking in certain places. Since I found some degree of difficulty in reading and understanding certain parts of the manuscript, the article needs some critical corrections and certain details which need to be incorporated. I do think that the manuscript contains important issues, interesting approaches, and techniques, which can lead to the understanding the role of integrins in testis fusion.

I suggest the authors to go for English corrections with a native speaker or a professional company.

Author Response

Reviewer comments:

The manuscript describes “Identification and expression of integrins during testes fusion in Spodoptera litura”.  The title of the manuscript is wider than the subject incorporated in it. The author tries to understand the role of integrins during testes fusion using molecular and microscopic techniques. The manuscript is well written, which I overall accept. The parameters chosen for strategy building and informative.

Response: We are very grateful for the reviewer’s support for our work, thank you for the valuable comments and questions. We change the title to ‘Identification and expression of integrins during testicular fusion in Spodoptera litura’

Introduction: Need to be improved with essential points to be highlighted. Please extend this part to give more glimpse for understanding the subject and provide latest references.

Response: Thank you! We have revised the Introduction part and supplemented the latest references. Over the past more than 20 years, the classical studies of integrins were very popular, so some references was not the latest, but those studies can still provide explanation and new ideas for our work.

Materials and methods: The structure and explanation of Section 2.1 can be improved.  Please provide all the information, unless you have already given an appropriate reference.

Response: Thank you! In this study, the artificial diet was prepared based on the reference (Chen et al., 2000, listed in line 90). We added the reference.

Chen Q.J., Li G.H., Pang Y. A simple artificial diet for mass rearing of some noctuid species. Entomol. Knowl. 2000;37:325–327.

Please correct spelling "Testes" to Testis through out the manuscript.

 Response: We have revised most ‘Testes’ to ‘Testis’, Thanks for your suggestion.

 Section 2.4: Do you try any other reference gene other than GAPDH? In case yes, provide some information. It’s always better to use minimum two reference genes. 

 Response: Thanks for your suggestion! In this study, we used the reference genes GAPDH based on the previous research “ Identification and Validation of Reference Genes for Gene Expression Analysis Using Quantitative PCR in Spodoptera litura (Lepidoptera: Noctuidae)” (Lu et al., 2013), which suggested GAPDH is a very stable reference gene for gene expression analysis using quantitative PCR at different developmental stages in the same tissue of S. litura. Our previous work on testis also confirmed GAPDH has stable expression during testis development.

Lu Y, Yuan M, Gao X, Kang T, Zhan S, Wan H, Li J. Identification and validation of reference genes for gene expression analysis using quantitative PCR in Spodoptera litura (Lepidoptera: Noctuidae). PLoS One. 2013, 8(7):e68059.

Results: Section 3.3 can be improved with more details(since there is enough to explain.

 Response: Thank you. We have made revisions accordingly. And we will look for help from ‘Language Editing’ service provided by MDPI.

Discussion part is very weak need to the reorganized because few parts of explanations are not connected and missing latest references. There should be a place where you need to explain the role of integrins explain mechanism that occur during developmental stage which is missing here.

Response: Thanks for your suggestion. We have made revisions in Discussion part and supplemented the latest references.

General comments:

Please double-check the manuscript for errors. In fact, the chosen subject must be made explicit in the manuscript. The content presented here is more theoretical. There are many places that need to be updated with the latest reference. I suggest the authors to go for English corrections with a native speaker or a professional company.

Response: Thanks for your suggestion! We have made revisions accordingly. The latest references were added accordingly. The English grammar has been improved accordingly in this version, and we will look for help from ‘Language Editing’ service provided by MDPI.

If possible, please provide the graphical abstract. This can help readers to understand better.

Response: Thanks for your suggestion, graphical abstract was added.

Reviewer 2 Report

The authors identified five α and five β integrin subunits closely associated with testicular fusion in Spodoptera litura. They first analyzed the mRNA expression profile in before and after the fusion of the testis using qRT-PCR, followed by studying the localization of these cell adhesion receptor integrins in the testes by immunohistochemistry. The manuscript provides insight into the molecular mechanism of the testicular fusion in lepidopteran insects. I think that the work is valuable. However, the following issues should be addressed in the next version.

1.     L37-38, it is better to delete the zebrafish citation.

2.     L52-54, If the molecular mechanism has previously been identified, but they need to be studied more, consider replacing this with something like “are poorly understood”. If the molecular mechanism has not been identified yet, consider using something like “are still unknown.”

3.     L134, how long each time?

4.     L175-178, these sentences are confusing, does not add anything and can be removed.

5.     L230-237, it is better to move these sentences to introduction part. 

6.     Figure 3, please double check y-axis â€¨of SllntaPDS3 and Sllnt β 5 !!

7.     L265-266, RT-qPCR can not be used to probe the physiological function, it can be used to explore differential expression patterns of the transcripts of interests between different tissue types or stages.

8.     L269-271,I believe you can remove this sentence “To determine…..”

9.     L280, remove [19].

10.  L284-285, “These results...” need more clarification or rewriting.

11.  L286-303, the description is interesting, but the authors only & over-statement the other reports. This paragraph is totally not a discussion, so please remove it. 

12.  It is better to find native English speaker to edit the M.S. 

It is better to find native English speaker to edit the M.S. 

Author Response

Conclusion:

The manuscript is interesting but strategic connectivity to the proposed hypothesis is lacking in certain places. Since I found some degree of difficulty in reading and understanding certain parts of the manuscript, the article needs some critical corrections and certain details which need to be incorporated. I do think that the manuscript contains important issues, interesting approaches, and techniques, which can lead to the understanding the role of integrins in testis fusion.

Response: We are grateful to the reviewer’s valuable comments and questions! We have made revisions accordingly.

Comments on the Quality of English Language

I suggest the authors to go for English corrections with a native speaker or a professional company.

Response: Thank you! The manuscript was edited by Letpub company before we submitted. This time, we will look for help from ‘Language Editing’ service provided by MDPI.

Comments and Suggestions for Authors

The authors identified five α and five β integrin subunits closely associated with testicular fusion in Spodoptera litura. They first analyzed the mRNA expression profile in before and after the fusion of the testis using qRT-PCR, followed by studying the localization of these cell adhesion receptor integrins in the testes by immunohistochemistry. The manuscript provides insight into the molecular mechanism of the testicular fusion in lepidopteran insects. I think that the work is valuable. However, the following issues should be addressed in the next version.

  1. L37-38, it is better to delete the zebrafish citation.

Response: Thanks for your suggestion! We have made revisions accordingly. 

  1. L52-54, If the molecular mechanism has previously been identified, but they need to be studied more, consider replacing this with something like “are poorly understood”. If the molecular mechanism has not been identified yet, consider using something like “are still unknown.”

Response: Thank you very much! We have made revisions accordingly. 

  1. L134, how long each time?

Response: The slides were dewaxed and rehydrated for 5 min, respectively. We added this information in the manuscript.

  1. L175-178, these sentences are confusing, does not add anything and can be removed.

Response: Thank you! We have made revisions accordingly. 

  1. L230-237, it is better to move these sentences to introduction part. 

Response: Thanks for your suggestion! We have moved these sentences to introduction part.  

  1. Figure 3, please double check y-axis of SllntaPDS3 and Sllnt β 5 !!

Response: Thank you! We have made revisions accordingly. 

  1. L265-266, RT-qPCR can not be used to probe the physiological function, it can be used to explore differential expression patterns of the transcripts of interests between different tissue types or stages.

Response: Thanks for your suggestion and we agree your comments. In this study, we only explored the expression patterns of integrins in testis by RT-qPCR. Currently, it is still difficulty to study the physiological function of five α integrins (α1, α2, α3, αPS1/PS2, αPS3) and two β integrins ( β1, β4) in S.litura. And studies show that there may be functional redundancy between integrins (Bui et al., 2019; Bell et al., 2008; Park et al., 2018).

Bui T, Rennhack J, Mok S, Ling C, Perez M, Roccamo J, Andrechek ER, Moraes C, Muller WJ. Functional Redundancy between β1 and β3 Integrin in Activating the IR/Akt/mTORC1 Signaling Axis to Promote ErbB2-Driven Breast Cancer. Cell Rep. 2019, 29(3):589-602.

Bell LV, Else KJ. Mechanisms of leucocyte recruitment to the inflamed large intestine: redundancy in integrin and addressin usage. Parasite Immunol. 2008, 30(3):163-170.

Park SH, Lee CW, Lee JH, Park JY, Roshandell M, Brennan CA, Choe KM. Requirement for and polarized localization of integrin proteins during Drosophila wound closure. Mol Biol Cell. 2018, 29(18):2137-2147.

  1. L269-271,I believe you can remove this sentence “To determine…..”

Response: Thank you! We have made revisions accordingly. 

  1. L280, remove [19].

Response: Thank you! We have removed the [19] accordingly.

  1. L284-285, “These results...” need more clarification or rewriting.

Response: Thank you! We have made revisions accordingly. 

  1. L286-303, the description is interesting, but the authors only & over-statement the other reports. This paragraph is totally not a discussion, so please remove it. 

Response: Thank you! We have removed the over-statement parts accordingly. 

  1. It is better to find native English speaker to edit the M.S. 

 Response: Thank you! We will make revisions with the help of ‘Language Editing’ service provided by MDPI.

Comments on the Quality of English Language

It is better to find native English speaker to edit the M.S. 

Response: Thank you. We will look for help from ‘Language Editing’ service provided by MDPI.